# Dynamical renormalization group analysis of $O(n)$ model in steady shear flow

Harukuni Ikeda[1*], and Hiroyoshi Nakano[2]

**1** Department of Physics, Gakushuin University, 1-5-1 Mejiro, Toshima-ku, Tokyo 171-8588, Japan
**2** Institute for Solid State Physics, University of Tokyo, 5-1-5, Kashiwanoha, Kashiwa 277-8581, Japan
* harukuni.ikeda@gakushuin.ac.jp

November 1, 2025

## Abstract

We study the critical behavior of the $O(n)$ model under steady shear flow using a dynamical renormalization group (RG) method. Incorporating the strong anisotropy in scaling ansatz, which has been neglected in earlier RG analyses, we identify a new stable Gaussian fixed point. This fixed point reproduces the anisotropic scaling of static and dynamical critical exponents for both non-conserved (Model A) and conserved (Model B) order parameters. Notably, the upper critical dimensions are $d_{up} = 2$ for the non-conserved order parameter (Model A) and $d_{up} = 0$ for the conserved order parameter (Model B), implying that the mean-field critical exponents are observed even in both $d = 2$ and $3$ dimensions. Furthermore, the scaling exponent of the order parameter is negative for all dimensions $d \geq 2$, indicating that shear flow stabilizes the long-range order associated with continuous symmetry breaking even in $d = 2$. In other words, the lower critical dimensions are $d_{low} < 2$ for both types of order parameters. This contrasts with equilibrium systems, where the Hohenberg—Mermin—Wagner theorem prohibits continuous symmetry breaking in $d = 2$.

# 1 Introduction

The $O(n)$ model is a cornerstone for studying critical phenomena and encompasses a wide range of phase transitions [1,2]. For $n = 1$, it describes transitions with discrete $Z_2$ symmetry breaking, such as Ising magnets and the liquid-gas transition. For $n \geq 2$, it captures continuous symmetry breaking, as observed in superfluid $^4$He and Heisenberg magnets. Moreover, this model provides insights into the behavior of liquid crystals. In equilibrium, the critical behavior of the $O(n)$ model is now well understood, thanks to advanced techniques in statistical mechanics, including renormalization group (RG) methods, exact solutions, and extensive numerical simulations [1,2]. However, far from equilibrium, our understanding of the $O(n)$ model is still under construction.

One common way to drive a system out of equilibrium is by applying external driving forces, such as shear [3–5]. In 1976, P. G. De Gennes theoretically investigated the scaling of the non-conserved order parameter near the second-order phase transition point in the steady shear flow [6]. His mean-field analysis predicts that the shear flow suppresses the critical fluctuations along the flow direction. As a consequence, at the critical point, the correlation function in the Fourier space for small wave vector $q$ exhibits the scaling $C(q) \sim |q_1|^{-2/3}$, where $q_1$ denotes the wave vector along the shear flow, instead of the standard Ornstein-Zernike like behavior $C(q) \sim |q|^{-2}$. Also, the dynamical scaling argument predicts that the correlation length $\xi$ and the relaxation time $\tau$ satisfy the scaling relation $\tau \sim \xi^{3/2}$ [6]. This scaling has been confirmed experimentally in nematic to smectic phase transition [7].

Subsequently, in 1979, Onuki and Kawasaki investigated the critical behavior of a conserved order parameter coupled to hydrodynamic variables in the presence of shear flow (Model H). Using a dynamical renormalization group (RG) approach, they predicted that the static critical exponents in three dimensions are described by the mean-field theory. This prediction was later confirmed in experiments on binary fluids in $d = 3$ [8,9]. Given that the upper critical dimension $d_{up}$ in equilibrium is $d_{up} = 4$, this result highlights a key distinction between equilibrium and nonequilibrium systems: shear flow can reduce the upper critical dimension.

The theoretical work of Onuki and Kawasaki has motivated numerous numerical investigations using the two-dimensional sheared Ising model. While the scaling $C(q) \sim |q_1|^{-2/3}$ has been explicitly confirmed by Monte Carlo simulations (Winter et al. [10], Angst et al. [11]), the value of critical exponent $\beta$ remains elusive. The simulations of the non-conserved sheared Ising model consistently yielded values below $1/2$, ranging from $\beta = 0.37$ (Winter et al. [10]) to $\beta = 0.39 \pm 0.01$ (Saracco and Gonnella [12]). However, the simulation of the conserved model by Saracco and Gonnella [13] showed a wider range of values for $\beta = 0.33 - 0.60$, some of which are close to $1/2$. Interestingly, the simulation of the two-dimensional sheared $O(2)$ model by Nakano et al. [14] obtained $\beta = 0.48$, a value much closer to the mean-field prediction. These conflicting results necessitate further investigation to resolve the controversy surrounding the mean-field characters in the sheared models.

More recently, the numerical simulation of the sheared $O(2)$ model [14] revealed the occurrence of long-range order associated with continuous symmetry breaking even in $d = 2$ dimension. For the $O(2)$ model, the order parameter fluctuations can be decomposed into phase and amplitude fluctuations. The phase fluctuations, which are soft modes, are referred

to as Nambu-Goldstone (NG) modes [14, 15]. It was found that shear flow not only modifies the critical fluctuations but also alters the fluctuations of the NG modes from the standard Ornstein-Zernike-like behavior, $C(\mathbf{q}) \sim |\mathbf{q}|^{-2}$, to a fractional scaling, $C(q_1) \sim |q_1|^{-2/3}$, thereby stabilizing the long-range order in two dimensions. This two-dimensional continuous symmetry breaking is remarkable because, in equilibrium systems, the lower critical dimension, $d_{\mathrm{low}}$, is 2, and the Hohenberg-Mermin-Wagner theorem prohibits continuous symmetry breaking in two dimensions [16, 17]. This reduction of the lower critical dimension further highlights a key distinction between equilibrium and nonequilibrium systems.

The reduction of $d_{\mathrm{up}}$ and $d_{\mathrm{low}}$ due to shear flow is a significant discovery in the study of non-equilibrium phase transitions. However, a complete theoretical understanding of this phenomenon is still lacking. The dynamical RG method offers a powerful approach to tackle this challenge. The aim of this paper is to propose that a stable fixed point can be obtained by correctly accounting for the anisotropy of the sheared system. The dynamical RG methods for anisotropic systems have been developed in several non-equilibrium systems, such as the directed percolation [18–21], growing interfaces [22–26], polar flocks [27–33], and coarsening dynamics subjected to the external field [34, 35] and the shear flow [36–39]. In this work, we apply the anisotropic dynamical RG formalism developed in these studies to the $O(n)$ model in the steady shear flow. We show that the anisotropic scaling ansatz leads to a new Gaussian fixed point, which is stable against shear. The upper critical dimension of the new fixed point is $d_{\mathrm{up}} = 2$ for a non-conserved order parameter (Model A) and $d_{\mathrm{up}} = 0$ for a conserved order parameter (Model B), meaning that the mean-field critical exponents are observed in $d = 2$ and 3.

The remainder of this paper is organized as follows. In Sec. 2, we investigate the model for the non-conservative order parameter (Model A). In Sec. 3, we investigate Model B in the steady shear. In Sec. 4, we conclude the work.

## 2 Model A

### 2.1 Settings

We consider the $d$-dimensional $O(n)$ model subjected to the steady shear [14, 15, 40]:

$$\dot{\phi}_a + \mathbf{v} \cdot \nabla \phi_a = D\nabla^2 \phi_a - \frac{\delta F[\vec{\phi}]}{\delta \phi_a} + \sqrt{2\Delta}\xi_a, \tag{1}$$

where the $n$-component vector $\vec{\phi}(\mathbf{x}, t) = \{\phi_1(\mathbf{x}, t), \cdots, \phi_n(\mathbf{x}, t)\}$ denotes the order parameter at position $\mathbf{x} = \{x_1, \cdots, x_d\}$ and time $t$,

$$F[\vec{\phi}] = \int d\mathbf{x} \left[ \frac{\varepsilon}{2} (\vec{\phi} \cdot \vec{\phi}) + \frac{u}{4} (\vec{\phi} \cdot \vec{\phi})^2 \right] \tag{2}$$

denotes the standard $\phi^4$ free-energy, and $\xi_a(\mathbf{x}, t)$ denotes the white noise whose mean and variance are given by

$$\langle \xi_a(\mathbf{x}, t) \rangle = 0,$$
$$\langle \xi_a(\mathbf{x}, t)\xi_b(\mathbf{x}', t') \rangle = \delta_{ab}\delta(\mathbf{x} - \mathbf{x}')\delta(t - t'). \tag{3}$$

The linear advection term in Eq. (1), $\mathbf{v} \cdot \nabla \phi_a$, represents the effect of the shear flow. We consider the simple shear flow along the $x_1$ axis with a constant gradient along the $x_2$ axis:

$$\mathbf{v} = \{\dot{\gamma} x_2, 0, \cdots, 0\}, \tag{4}$$

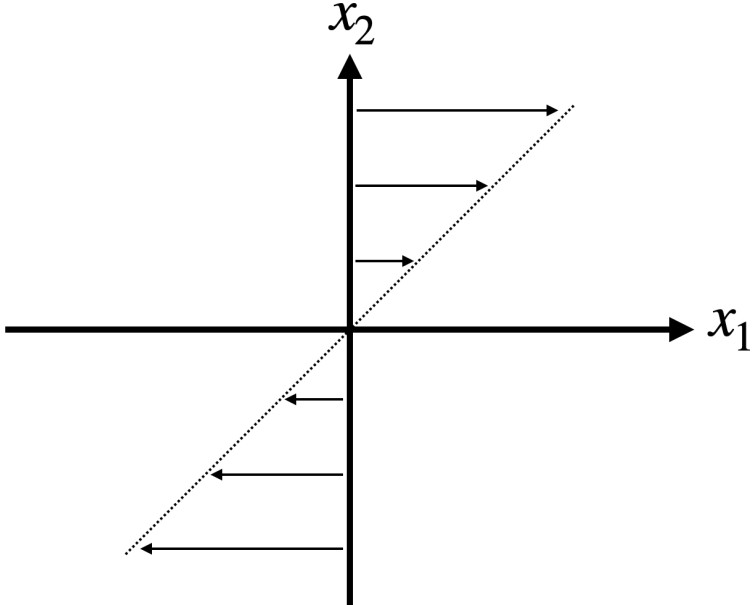

Figure 1: Schematic picture of the shear flow.

where $\dot{\gamma}$ denotes the shear rate, see Fig. 1.

The $O(n)$ model for $n = 1$ describes the Ising universality class. In previous works, the Ising model in shear has been studied extensively [10–12, 41, 42]. In particular, for the infinitely large shear rate $\dot{\gamma} \to \infty$, the mean-field approximation becomes exact, which enables to solve the model analytically [42]. However, numerical simulations have produced the conflicting results, as summarized in Introduction [10–12, 41, 42]. The $O(n)$ model in shear with $n = 2$ also has been studied extensively [14,15,40]. A recent numerical simulation has demonstrated that the model undergoes the continuous symmetry breaking even in $d = 2$. Interestingly, the critical exponents of the transition agree with those of the mean-field prediction [15, 40].

## 2.2 Renormalization Group flow equations

To investigate the large spatiotemporal behavior of the model near the critical point $\varepsilon = 0$, we consider the anisotropic scaling transformations [24, 28, 37]:

$$x_1 = l^\zeta x_1', \; \boldsymbol{x}_\perp = l\boldsymbol{x}_\perp', \; t = l^z t', \; \phi_a = l^\chi \phi_a', \tag{5}$$

where $x_1$ and $\boldsymbol{x}_\perp = \{x_2, \cdots, x_d\}$ respectively denote the coordinates parallel and perpendicular to the flow direction. The above scaling is tantamount to assuming that the typical size of the fluctuations grows anisotropically as $x_1 \sim t^{\zeta/z}$ and $\boldsymbol{x}_\perp \sim t^{1/z}$, see Fig. 2. Substituting the scaling relations (5) into the equation of motion (1) and dividing both sides by $l^{\chi-z}$, we get

$$\dot{\phi}_a' + l^{z+1-\zeta} \dot{\gamma} x_2' \partial_1' \phi_a'$$
$$= l^{z-2\zeta} D_\parallel (\partial_1')^2 \phi_a' + l^{z-2} D_\perp (\nabla_\perp')^2 \phi_a' - l^z \varepsilon \phi_a' - l^{z+2\chi} u(\vec{\phi}' \cdot \vec{\phi}')\phi_a' + l^{\frac{z-2\chi-(d-1+\zeta)}{2}} \sqrt{2\Delta} \xi_a', \tag{6}$$

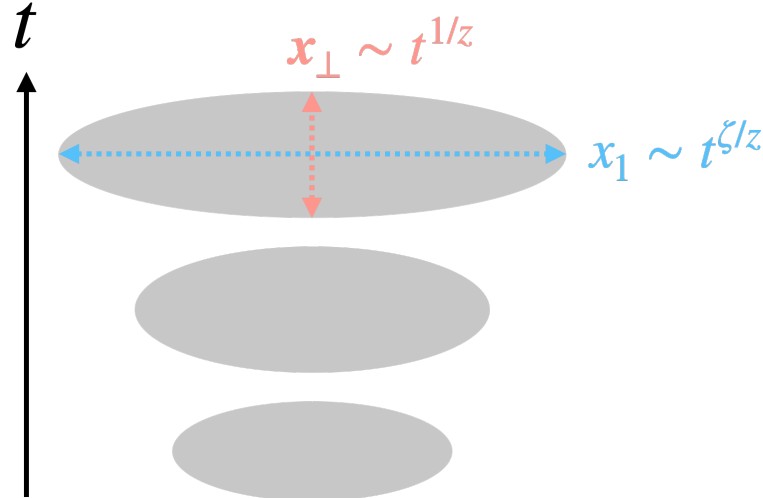

$$x_\perp \sim t^{1/z}$$
$$x_1 \sim t^{\zeta/z}$$

Figure 2: Schematic picture of the typical size of the critical fluctuation.

where $D_\parallel = D_\perp = D$. The above equation implies that under the scaling transformations (5), the coefficients of the equation of motion (1) are transformed to

$$
\begin{aligned}
\dot{\gamma}' &= l^{z+1-\zeta}\dot{\gamma}, \\
D_\parallel' &= l^{z-2\zeta}D_\parallel, \\
D_\perp' &= l^{z-2}D_\perp, \\
\varepsilon' &= l^z \varepsilon, \\
u' &= l^{z+2\chi}u, \\
\Delta' &= l^{z-2\chi-(d-1+\zeta)}\Delta,
\end{aligned}
\tag{7}
$$

leading to the RG flow equations:

$$\left.\frac{d\dot{\gamma}}{dl}\right|_{l=1} = (z+1-\zeta)\dot{\gamma}, \tag{8}$$

$$\left.\frac{dD_\parallel}{dl}\right|_{l=1} = (z-2\zeta)D_\parallel, \tag{9}$$

$$\left.\frac{dD_\perp}{dl}\right|_{l=1} = (z-2)D_\perp, \tag{10}$$

$$\left.\frac{d\Delta}{dl}\right|_{l=1} = [z-2\chi-(d-1+\zeta)]\Delta, \tag{11}$$

$$\left.\frac{d\varepsilon}{dl}\right|_{l=1} = z\varepsilon, \tag{12}$$

$$\left.\frac{du}{dl}\right|_{l=1} = (z+2\chi)u. \tag{13}$$

The equations are derived by a naive scaling argument, which is tantamount to neglecting the graphical corrections [43]. The equations can be applied only near the Gaussian fixed point. The stability of the Gaussian fixed point can be discussed by the RG flow equation of the non-linear term, Eq. (13), as we will discuss below.

## 2.3  Gaussian fixed point without shear

Before investigating the critical behavior under shear flow, we review the RG analysis for the equilibrium case ($\dot{\gamma} = 0$) of a non-conserved order parameter (Model A) to establish a baseline for comparison. For $\dot{\gamma} = 0$, Eq. (8) is automatically satisfied. We assume that the transport coefficients and the noise strength are scale-invariant, which requires

$$\frac{dD_\parallel}{dl}\bigg|_{l=1} = \frac{dD_\perp}{dl}\bigg|_{l=1} = \frac{d\Delta}{dl}\bigg|_{l=1} = 0, \tag{14}$$

leading to the scaling relations

$$z - 2\zeta = z - 2 = z - 2\chi - (d - 1 + \zeta) = 0. \tag{15}$$

This can be solved easily, and we get

$$\zeta = 1, \quad z = 2, \quad \chi = \frac{2 - d}{2}. \tag{16}$$

The results are consistent with those of the Gaussian fixed point in equilibrium [1].

   First, we consider the upper critical dimension, $d_{\text{up}}$. Above this dimension, the critical behavior of the system is captured by mean-field theory, resulting in mean-field critical exponents. To determine the upper critical dimensions, $d_{\text{up}}$, we analyze the stability of the Gaussian fixed point against the non-linear term $u$. By substituting the exponents (16) into the RG flow equation for the non-linear term (13), we have

$$\frac{du}{dl}\bigg|_{l=1} = (4 - d)u, \tag{17}$$

leading to $u \sim l^{4-d}$. For $d < 4$, the coefficient diverges $u \to \infty$ in the thermodynamic limit $l \to \infty$, which destabilizes the Gaussian fixed point. On the contrary, for $d > 4$, the coefficient vanishes $u \to 0$, meaning that the Gaussian fixed point is stable. Therefore, the upper critical dimension $d_{\text{up}}$ is

$$d_{\text{up}} = 4. \tag{18}$$

   Note that $u$ is a dangerously irrelevant variable above the upper critical dimensions $d > d_{\text{up}}$ [1]. In other words, $u$ is irrelevant but still affects the scaling. To understand this, let us write the scaling law of $\phi_a$ in the ordered phase $\varepsilon < 0$ with the variable $u$ explicitly included as

$$\phi_a(\varepsilon, u) = l^\chi \phi_a(l^z \varepsilon, l^{z+2\chi} u). \tag{19}$$

Setting $l = |\varepsilon|^{-1/z}$, we get

$$\phi_a(\varepsilon, u) = |\varepsilon|^{-\chi/z} \phi_a(-1, |\varepsilon|^{-(z+2\chi)/z} u). \tag{20}$$

One may naively expect that $\phi_a(-1, u)$ is analytic in $u$ and $\phi_a(-1, u) \sim u^0$ for $u \ll 1$, leading to $\langle \phi_a \rangle \sim |\varepsilon|^\beta$ with $\beta = -\chi/z = d/4$. However, in fact, $\phi_a(-1, u)$ is not analytic in $u$ for $d > d_{\text{up}}$. This can be seen from the saddle point equation of the free-energy (2),

$$\frac{\delta F}{\delta \phi_a} = \varepsilon \phi_a + u(\vec{\phi} \cdot \vec{\phi})\phi_a = 0, \tag{21}$$

which implies $\phi_a(-1, u) \sim u^{-1/2}$ for $u \ll 1$. Consequently, we have $\phi_a(-1, u) \sim u^{-1/2}$ instead of $\phi_a(-1, u) \sim u^0$, and thus

$$\phi_a(\varepsilon, u) \sim |\varepsilon|^{-\chi/z} \left( |\varepsilon|^{-(z+2\chi)/z} u \right)^{-1/2} \sim |\varepsilon|^{1/2}. \tag{22}$$

for $|\varepsilon| \ll 1$. Therefore, the correct critical exponent above $d_{\text{up}}$ is

$$\beta = \frac{1}{2}. \tag{23}$$

For more detailed discussions, see, for instance, Sec. 4. 2 in Ref. [1].

Next, we study the lower critical dimension $d_{\text{low}}$. It is defined as the dimension below which the long-range order associated with continuous symmetry breaking is destroyed by fluctuations. For the $O(n)$ model for $n \geq 2$, consider the ordered phase ($\varepsilon < 0$) where the order parameter is oriented along the $a = 1$ direction, such as $\langle \phi_a \rangle = (-\varepsilon/u)^{1/2} \delta_{a1}$. The fluctuations of the order parameter perpendicular to this direction, denoted by $\delta \phi_a$ ($a \neq 1$), correspond to the NG modes [44, 45]. The NG modes become divergent below the lower critical dimension $d_{\text{low}}$, destroying the long-range order. To calculate $d_{\text{low}}$, we observe the scaling for $\delta \phi_a$ ($a \neq 1$) [28]. They follow the dynamics $\delta \dot{\phi}_a = D \nabla^2 \delta \phi_a + \sqrt{2\Delta} \xi_a$ for $a \neq 1$. A similar scaling analysis as above leads to

$$\left\langle \delta \phi_a^2 \right\rangle \sim l^{2\chi}. \tag{24}$$

If $\chi > 0$, the fluctuations diverge in the thermodynamic limit $l \to \infty$, meaning that a necessary condition for the continuous symmetry breaking is $\chi = (2-d)/2 < 0$, or equivalently, $d > 2$. Therefore, the lower critical dimension is

$$d_{\text{low}} = 2, \tag{25}$$

which is consistent with the Hohenberg–Mermin–Wagner theorem [16, 17].

Finally, we discuss that the Gaussian fixed point is destabilized by the shear in any $d$. Substituting (16) into the RG flow equation of $\dot{\gamma}$ (8), we get

$$\left. \frac{d\dot{\gamma}}{dl} \right|_{l=1} = 2\dot{\gamma}, \tag{26}$$

leading to $\dot{\gamma} \sim l^2$. This means that the shear rate $\dot{\gamma}$ is a relevant variable, which destabilizes the equilibrium Gaussian fixed point in any spatial dimension $d$. Therefore, we need to seek a new stable fixed point, which will be discussed in the next section.

## 2.4 Gaussian fixed point with shear

Now, we investigate the model with the finite shear rate $\dot{\gamma} \neq 0$. For this purpose, we observe the scaling behaviors of the shear rate $\dot{\gamma}$, the transport coefficients $D_\parallel$, $D_\perp$, and the strength of the noise $\Delta$, whose RG flow equations are given by (8)-(11). To determine the three independent exponents, $\zeta$, $z$, and $\chi$, we need three independent conditions. For instance, the scale invariance of $D_\parallel$, $D_\perp$, and $\Delta$ leads to the equilibrium Gaussian fixed point, which is unstable against shear, as discussed in the previous subsection. Another physically meaningful solution can be obtained by requiring the scale invariance of $\dot{\gamma}$, $D_\perp$, and $\Delta$:

$$\left. \frac{d\dot{\gamma}}{dl} \right|_{l=1} = \left. \frac{dD_\perp}{dl} \right|_{l=1} = \left. \frac{d\Delta}{dl} \right|_{l=1} = 0, \tag{27}$$

or equivalently,

$$z + 1 - \zeta = z - 2 = z - 2\chi - (d - 1 + \zeta) = 0, \tag{28}$$

leading to

$$\zeta = 3, \quad z = 2, \quad \chi = -\frac{d}{2}. \tag{29}$$

Any other choices yield unphysical results [1]. The scaling $x_\parallel \sim l^\zeta$ and $t \sim l^z$ imply the scaling relation between the relaxation time $\tau$ and correlation length along the flow direction $\xi_\parallel$: $\tau \sim \xi_\parallel^{2/3}$, which is consistent with previous studies [6,7,42]. Substituting the above exponents into Eq. (9), we get

$$\left.\frac{dD_\parallel}{dl}\right|_{l=1} = -4D_\parallel, \tag{30}$$

leading to $D_\parallel \sim l^{-4}$, which vanishes in the thermodynamic limit $l \to \infty$. Therefore, the diffusion term parallel to the flow direction is irrelevant.

The upper critical dimension $d_{\text{up}}$ is calculated by observing the RG flow equation for the non-linear term (13):

$$\left.\frac{du}{dl}\right|_{l=1} = (2-d)u, \tag{31}$$

leading to $u \sim l^{2-d}$. For $d > 2$, $u$ vanishes in the thermodynamic limit $l \to \infty$, meaning that the upper critical dimension is

$$d_{\text{up}} = 2, \tag{32}$$

This suggests that the mean-field critical exponents are observed in physical dimensions $d = 2$ and 3. Note that, similar to the equilibrium case, $u$ remains a dangerously irrelevant variable. Assuming the same scaling behavior as in equilibrium, namely $\phi_a(-1, u) \sim u^{-1/2}$ instead of $\phi_a(-1, u) \sim u^0$, we obtain

$$\phi_a(\varepsilon, u) \sim |\varepsilon|^{-\chi/z} \left(|\varepsilon|^{-(z+2\chi)/z} u\right)^{-1/2} \sim |\varepsilon|^{1/2}. \tag{33}$$

for $|\varepsilon| \ll 1$. Therefore, the critical exponent $\beta$ above $d_{\text{up}} = 2$ is

$$\beta = \frac{1}{2}, \tag{34}$$

which is consistent with theoretical studies in the limit of large shear rate [42]. However, as mentioned in the Introduction, previous numerical studies in two dimensions have often reported values of $\beta$ that deviate from $1/2$ [10–12,41,42]. This discrepancy may be attributed to the presence of logarithmic corrections at the upper critical dimension [1]. Future numerical work should investigate this possibility.

The lower critical dimension $d_{\text{low}}$ is also varied in the presence of shear flow. Indeed, $\chi < 0$ for any $d > 0$, which means that $\langle \delta \phi_a^2 \rangle \sim l^{2\chi} \to 0$, and then the lower critical dimension $d_{\text{low}}$ turns out to be

$$d_{\text{low}} = 0. \tag{35}$$

In particular, the continuous symmetry breaking can occur even in $d = 2$, which is prohibited in equilibrium by the Hohenberg–Mermin–Wagner theorem [16,17]. The result is consistent with a recent numerical simulation for the $O(2)$ model in $d = 2$ [14].

---

[1]The scale invariance of $\dot{\gamma}$, $D_\parallel$, and $\Delta$ leads to a negative value of the dynamical critical exponent $z = -2$, which is unphysical. The scale invariance of $\dot{\gamma}$, $D_\parallel$, and $D_\perp$ leads to $z + 1 - \zeta = z - 2\zeta = z - 2 = 0$, which does not have the solution.

## 2.5 Correlation functions

The presence of shear flow significantly alters the behavior of correlation functions, both for the critical fluctuations and the NG modes. We first investigate the correlation function just above the critical point ($\varepsilon > 0$). The scaling behaviors (5) and (7) lead to

$$
\begin{aligned}
C(x_1, \boldsymbol{x}_\perp, \varepsilon) &= \left\langle \vec{\phi}(x_1, \boldsymbol{x}_\perp) \cdot \vec{\phi}(0,0) \right\rangle \\
&= l^{2\chi} C(l^{-\zeta} x_1, l^{-1} \boldsymbol{x}_\perp, l^z \varepsilon).
\end{aligned}
\tag{36}
$$

In the Fourier space, we get

$$
\begin{aligned}
C(q_1, \boldsymbol{q}_\perp, \varepsilon) &= \int d\boldsymbol{x} e^{i(q_1 x_1 + \boldsymbol{q}_\perp \cdot \boldsymbol{x}_\perp)} C(x_1, \boldsymbol{x}_\perp, \varepsilon) \\
&= l^{2\chi + \zeta + d - 1} C(l^\zeta q_1, l \boldsymbol{q}_\perp, l^z \varepsilon) = l^z C(l^\zeta q_1, l \boldsymbol{q}_\perp, l^z \varepsilon)
\end{aligned}
\tag{37}
$$

Substituting $l = \varepsilon^{-1/z}$, we get

$$
C(q_1, \boldsymbol{q}_\perp, \varepsilon) = \varepsilon^{-1} C(\varepsilon^{-\nu_\parallel} q_1, \varepsilon^{-\nu_\perp} \boldsymbol{q}_\perp, 1)
\tag{38}
$$

with

$$
\nu_\parallel = \frac{3}{2}, \qquad\qquad \nu_\perp = \frac{1}{2}.
\tag{39}
$$

The scaling form (38) implies that the correlation lengths parallel and perpendicular to the flow direction, $\xi_\parallel$ and $\xi_\perp$, diverge as

$$
\xi_\parallel \sim \varepsilon^{-\nu_\parallel}, \qquad\qquad \xi_\perp \sim \varepsilon^{-\nu_\perp}.
\tag{40}
$$

The exponents are consistent with the previous theoretical and numerical simulations for the Ising-model [11, 42], and $O(n)$ model [14, 15, 40] in shear. Repeating the similar scaling analysis, we get

$$
\begin{aligned}
C(q_1, \boldsymbol{0}, 0) &\sim |q_1|^{-2/3}, & |q_1| &\ll 1, & (41) \\
C(0, \boldsymbol{q}_\perp, 0) &\sim |\boldsymbol{q}_\perp|^{-2}, & |\boldsymbol{q}_\perp| &\ll 1, & (42) \\
C(0, 0, \varepsilon) &\sim \varepsilon^{-1}, & \varepsilon &\ll 1, & (43)
\end{aligned}
$$

implying that the correlation function can be expanded as

$$
C(q_1, \boldsymbol{q}_\perp, \varepsilon) = \left[ c_1 \varepsilon + c_2 |q_1|^{2/3} + c_3 |\boldsymbol{q}_\perp|^2 + \cdots \right]^{-1},
\tag{44}
$$

where $c_1$, $c_2$, and $c_3$ are some constants. This is consistent with the previous studies for the Ising model in shear [11, 42].

We proceed to the analysis of the NG mode. Following the deviation of Eq. (24), we consider the ordered phase ($\varepsilon < 0$) where the order parameter is oriented along the $a = 1$ direction, such as $\langle \phi_a \rangle = (-\varepsilon/u)^{1/2} \delta_{a1}$. The NG mode, $\delta\phi_a$ ($a \neq 1$), is governed by the equation $\delta\dot{\phi}_a + \dot{\gamma} x_2 \partial_1 \phi_a = D\nabla^2 \delta\phi_a + \sqrt{2\Delta} \xi_a$, and then a similar scaling analysis as above leads to $C(q_1, \boldsymbol{q}_\perp, \varepsilon) \approx (c_2 |q_1|^{2/3} + c_3 |\boldsymbol{q}_\perp|^2)^{-1}$ for $\varepsilon < 0$. The result is consistent with the recent numerical simulation [14] and linear analysis [15].

# 3 Model B

Model B is a simple variation of Model A, which describes the dynamics of the conserved order parameter, such as density in the phase separation [46]. In equilibrium, both models share the same static critical exponents, since their values are determined solely by the free energy and are independent of the specific dynamics. Their dynamical critical exponents differ due to their distinct dynamics [1, 46]. In contrast, we show that under uniform shear flow, even the static critical exponents depend on the type of dynamics, leading to different values for Model A and Model B. Furthermore, we find that critical fluctuations are more strongly suppressed in Model B compared to Model A, leading to smaller values for the critical dimensions, $d_{\text{low}}$ and $d_{\text{up}}$.

## 3.1 Settings

We consider the following equation of motion for the conserved-order parameter in the steady shear flow (Model B) [46, 47]:

$$\dot{\phi}_a + \boldsymbol{v} \cdot \nabla \phi_a = -\nabla^2 \left[ D\nabla^2 \phi_a + \frac{\delta F[\vec{\phi}]}{\delta \phi_a} \right] + \sqrt{2\Delta} \nabla \cdot \boldsymbol{\xi}_a, \tag{45}$$

where $F[\vec{\phi}]$ denotes the free-energy (2), $\boldsymbol{v}$ denotes the velocity of the shear flow (4), and $\boldsymbol{\xi}_a = \{\xi_{a,1}, \cdots, \xi_{a,d}\}$ denotes the white noise whose mean and variance are given by

$$\begin{aligned} \left\langle \xi_{a,i}(\boldsymbol{x}, t) \right\rangle &= 0, \\ \left\langle \xi_{a,i}(\boldsymbol{x}, t)\xi_{b,j}(\boldsymbol{x}', t') \right\rangle &= \delta_{ab}\delta_{ij}\delta(\boldsymbol{x} - \boldsymbol{x}')\delta(t - t'). \end{aligned} \tag{46}$$

## 3.2 Renormalization Group flow equations

To investigate the large spatiotemporal behavior of the model, we consider the anisotropic scaling transformations (5). As we will see later, the anisotropic parameter is $\zeta > 1$, which enables the following approximation:

$$\nabla = l^{-\zeta}\partial'_1 \boldsymbol{e}_1 + l^{-1}\nabla'_\perp \approx l^{-1}\nabla'_\perp, \tag{47}$$

where $\boldsymbol{e}_1$ denotes the unit vector along $x_1$. After some manipulations, we get

$$\begin{aligned} &\dot{\phi}'_a + l^{z+1-\zeta}\dot{\gamma}x'_2\partial'_1\phi'_a \\ &\approx l^{z-4}D_\perp(\nabla'_\perp)^4\phi'_a - l^{z-2}(\nabla'_\perp)^2\varepsilon\phi'_a - l^{z+2\chi-2}(\nabla'_\perp)^2 u(\vec{\phi}'\cdot\vec{\phi}')\phi'_a + l^{\frac{z-2\chi-(d-1+\zeta)-2}{2}}\sqrt{2\Delta}\nabla'_\perp \cdot \boldsymbol{\xi}'_a, \end{aligned} \tag{48}$$

leading to the RG flow equations [1]:

$$\left. \frac{d\dot{\gamma}}{dl} \right|_{l=1} = (z + 1 - \zeta)\dot{\gamma}, \tag{49}$$

$$\left. \frac{dD_\perp}{dl} \right|_{l=1} = (z - 4)D_\perp, \tag{50}$$

$$\left. \frac{d\Delta}{dl} \right|_{l=1} = [z - 2\chi - (d - 1 + \zeta) - 2]\Delta, \tag{51}$$

$$\left. \frac{d\varepsilon}{dl} \right|_{l=1} = (z - 2)\varepsilon, \tag{52}$$

$$\left. \frac{du}{dl} \right|_{l=1} = (z + 2\chi - 2)u. \tag{53}$$

The diffusion parallel to the flow direction $D_\parallel$ does not appear since the spatial derivative along that direction was already dropped in the approximation (47).

### 3.3 Gaussian fixed point with shear

As in the case of Model A, we require the scale invariance of $\dot\gamma$, $D_\perp$, and $\Delta$:

$$\left.\frac{d\dot\gamma}{dl}\right|_{l=1} = \left.\frac{dD_\perp}{dl}\right|_{l=1} = \left.\frac{d\Delta}{dl}\right|_{l=1} = 0, \tag{54}$$

leading to

$$z + 1 - \zeta = z - 4 = z - 2\chi - (d - 1 + \zeta) - 2 = 0. \tag{55}$$

Solving the above scaling relations, we get the following critical exponents:

$$\zeta = 5, \quad z = 4, \quad \chi = -\frac{d+2}{2}. \tag{56}$$

The anisotropic exponent satisfies $\zeta > 1$, which justifies the approximation Eq. (47). The RG flow equation for the non-linear term (53) is

$$\left.\frac{du}{dl}\right|_{l=1} = -du, \tag{57}$$

leading to $u \sim l^{-d}$. For $d > 0$, $u \to 0$ in the thermodynamic limit $l \to \infty$, meaning that the upper critical dimension is

$$d_{\text{up}} = 0. \tag{58}$$

Therefore, the mean-field critical exponents are observed for any $d > 0$. Furthermore, the scaling exponent of the order parameter $\chi$ becomes negative for $d > d_{\text{low}}$ with

$$d_{\text{low}} = -2. \tag{59}$$

In particular, the continuous symmetry breaking can occur in $d = 2$, as in the case of Model A with shear.

### 3.4 Correlation function

The scaling behaviors, $\phi \sim l^\chi$, $x_1 \sim l^\zeta$, $x_\perp \sim l$, and $\varepsilon \sim l^{-(z-2)}$, lead to

$$\begin{aligned}
C(x_1, x_\perp, \varepsilon) &= \left\langle \vec\phi(x_1, x_\perp) \cdot \vec\phi(0,0) \right\rangle \\
&= l^{2\chi} C(l^{-\zeta} x_1, l^{-1} x_\perp, l^{z-2}\varepsilon).
\end{aligned} \tag{60}$$

In the Fourier space, we get

$$\begin{aligned}
C(q_1, q_\perp, \varepsilon) &= \int dx \, e^{i(q_1 x_1 + q_\perp \cdot x_\perp)} C(x_1, x_\perp, \varepsilon) \\
&= l^{2\chi + \zeta + d - 1} C(l^\zeta q_1, l q_\perp, l^{z-2}\varepsilon) \\
&= l^{z-2} C(l^\zeta q_1, l q_\perp, l^{z-2}\varepsilon).
\end{aligned} \tag{61}$$

Substituting $l = \varepsilon^{-1/(z-2)}$, we get

$$C(q_1, q_\perp, \varepsilon) = \varepsilon^{-1} C(\varepsilon^{-\nu_\parallel} q_1, \varepsilon^{-\nu_\perp} q_\perp, 1) \tag{62}$$

Table 1: Lower and upper critical dimensions, and critical exponents in equilibrium and shear flow. Note that the shear flow can be defined only in $d \geq 2$.

|  | in equilibrium | | in shear flow | |
|---|---|---|---|---|
|  | Model A | Model B | Model A | Model B |
| $d_{\mathrm{low}}$ | 2 | 2 | 0 | -2 |
| $d_{\mathrm{up}}$ | 4 | 4 | 2 | 0 |
| $\zeta$ | 1 | 1 | 3 | 5 |
| $z$ | 2 | 4 | 2 | 4 |
| $\chi$ | (2-d)/2 | (2-d)/2 | -d/2 | -(d+2)/2 |
| $\nu_{\parallel}$ | 1/2 | 1/2 | 3/2 | 5/2 |
| $\nu_{\perp}$ | 1/2 | 1/2 | 1/2 | 1/2 |

with

$$\nu_{\parallel} = \frac{5}{2}, \quad \nu_{\perp} = \frac{1}{2}. \tag{63}$$

The exponents are consistent with the previous mean-field analysis for the model-H in OK79 [48–51]. This is a reasonable result, because the order parameter and hydrodynamic momentum of the model-H decouple at the level of the linear analysis, leading to the same equation of motion as that of Model B [48]. Also, from Eq. (61), we get

$$C(q_1, \mathbf{0}, 0) \sim |q_1|^{-2/5}, \qquad\qquad |q_1| \ll 1, \tag{64}$$

$$C(0, \mathbf{q}_{\perp}, 0) \sim |\mathbf{q}_{\perp}|^{-2}, \qquad\qquad |\mathbf{q}_{\perp}| \ll 1, \tag{65}$$

$$C(0, 0, \varepsilon) \sim \varepsilon^{-1}, \qquad\qquad \varepsilon \ll 1, \tag{66}$$

implying the following expansion:

$$C(q_1, \mathbf{q}_{\perp}, \varepsilon) = \left[ c_1 \varepsilon + c_2 |q_1|^{2/5} + c_3 |\mathbf{q}_{\perp}|^2 + \cdots \right]^{-1}, \tag{67}$$

which are again consistent with the theoretical prediction in OK79 [48]. The correction along the flow direction $C(q_1, \mathbf{0}, 0) \sim |q_1|^{-2/5}$, Eq. (64), is much smaller than that for Model A $C(q_1, \mathbf{0}, 0) \sim |q_1|^{-2/3}$, Eq (41), meaning that the critical fluctuations are more strongly suppressed than those of Model A. This observation supports that Model B has smaller critical dimensions, $d_{\mathrm{low}} = -2$ and $d_{\mathrm{up}} = 0$, compared to Model A, with $d_{\mathrm{low}} = 0$ and $d_{\mathrm{up}} = 2$.

## 4 Summary and discussions

In this work, we investigated the $O(n)$ model subjected to the steady shear flow for both non-conserved (Model A) and conserved (Model B) order parameters. Using the dynamical RG analysis incorporating the anisotropic scaling, we identify a new Gaussian fixed point that is stable under the shear flow. Table 1 summarizes the critical dimensions and exponents corresponding to this fixed point. In addition, we calculated the scaling behaviors of the correlation functions near this Gaussian fixed point. The correction in the Fourier space behaves as $C(q_1) \sim |q_1|^{-a}$, where $q_1$ denotes the wave vector along the flow direction. The exponent $a$ is smaller than its equilibrium value of 2 ($a = 2/3$ for Model A and $a = 2/5$ for Model B), indicating that the shear flow suppresses the critical fluctuations along the flow direction. This suppression is consistent with the reduction of the critical dimensions, $d_{\mathrm{low}}$ and $d_{\mathrm{up}}$.

We found that $d_{\mathrm{up}} = 2$ for Model A and $d_{\mathrm{up}} = 0$ for Model B [2]. This means that for both Model A and B, the critical exponents agree with those of the mean-field predictions in $d = 2$ and 3. As mentioned in the Introduction, previous numerical studies in two-dimensional non-conserved Ising model have often reported values of $\beta$ that deviate from the mean-field critical exponent $1/2$ [10–12, 41, 42]. This discrepancy may arise from the presence of logarithmic corrections at $d_{\mathrm{up}}$ [1]. For the conservative dynamics, a recent numerical result for the Ising model shows a clear deviation from the mean-field prediction [13]. We speculate that this discrepancy comes from the strong finite-size effect caused by the large anisotropy exponent $\zeta = 5$. The large anisotropy exponent, $\zeta = 5$, implies that the system size needs to be scaled anisotropically as $(L_\parallel, L_\perp) = (l^5, l)$ to allow for sufficient development of critical fluctuations, where $L_\parallel$ and $L_\perp$ are the linear sizes of the system parallel and perpendicular to the flow direction, respectively. Therefore, if we consider a system with $L_\perp \approx 10$, $L_\parallel \approx 10^5$ is required to accurately estimate the critical exponents. The standard finite-size scaling analysis, as used in Ref. [13], would fail to capture such extreme anisotropy. It is interesting future work to develop methods to precisely and efficiently calculate the critical exponents of those systems by using numerical simulations.

Finally, we comment on the relation between the present analysis and the earlier work by Onuki and Kawasaki (OK79) [48]. In OK79, the authors studied Model H under shear flow by assuming isotropic scaling, while introducing anisotropy through wave-number–dependent renormalized transport coefficients. Their renormalization-group equations [Eqs. (4.24)–(4.25) in OK79] depend on both amplitude and angle of wave vector, and the resulting fixed point corresponds to the infinite shear-rate limit. In contrast, the present study focuses on Models A and B using an explicitly anisotropic-scaling framework. Because the two analyses are based on different theoretical formalisms, their scaling predictions are not directly comparable. A promising direction for future work would be to apply OK79's approach to Models A and B, or conversely, to extend the present anisotropic-scaling analysis to Model H, which could help bridge these complementary frameworks.

## Acknowledgements

We thank H. Tasaki, Y. Kuroda, M. Hongo, and S. Sasa for useful discussions. The authors thank YITP at Kyoto University and RIKEN iTHEMS. Discussions during the workshop (YITP-T-24-04) on "Advances in Fluctuating Hydrodynamics: Bridging the Micro and Macro Scales" were useful in completing this work.

**Funding information** This project has received JSPS KAKENHI Grant Numbers 23K13031.

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
