# Peer review of "Dynamical renormalization group analysis of $O(n)$ model in steady shear flow"

_SciPost Physics_

## Round 1 · Author Response

Dear Editor,

We are pleased to resubmit our manuscript in response to the referees’ reports. We sincerely thank both referees for their careful reading and positive evaluations of our work. The first referee had no particular requests for changes, while the second referee suggested that we add more discussions of the previous work by Onuki and Kawasaki (OK79). Below, we provide our replies to the comments and describe the major changes made in the revised version.

We hope and believe that you will find the revised manuscript suitable for publication in SciPost Physics.

Thank you very much for your kind consideration.

Sincerely yours, Harukuni Ikeda, and Hiroyoshi Nakano.

Reply to Report #1

[REFEREE] This is a nice paper, which faces an open problem in a simple and yet sharp way. By doing scaling analysis in an anisotropic setup around the Gaussian fixed point, a new set of critical exponents is found for the case of linear shear in Model A and Model B: the nonlinearity is (dangerously) irrelevant for d≥2 and d≥0 respectively, so that everything can be done at tree level, which is surprising, but nice. The paper is well-written and it uses standard well-known scaling arguments in a simple way, hence I have no particular requests of any changes.

[REPLY] We thank the referee for careful reading and positive evaluation of our manuscript.

Reply to Report #2

[REFEREE] This manuscript introduces a novel anisotropic scaling approach for the Gaussian fixed point in O(N) models under shear. Using this scaling, the Gaussian fixed point becomes stable in both d=2 and d=3 dimensions for the dynamic models referred to as Model A and Model B. I believe this is an important observation that should be compared with experimental results and simulations, and it merits publication.

[REPLY] We thank the referee for careful reading and positive evaluation of our manuscript.

[REFEREE] The author asserts that the nontrivial fixed point proposed by Onuki and Kawasaki (OK), which is unstable under shear, does not adequately describe dynamical critical phenomena. My understanding is that, while OK acknowledged the instability of their fixed point with respect to shear, they conducted analyses around the fixed point to discuss scaling behavior under such conditions. Therefore, I feel it is necessary to provide a more detailed discussion of how and to what extent the analysis by OK is rendered invalid.

Requested changes 1- More comments on the previous work by Onuki and Kawasaki

[REPLY] We thank the referee for this helpful comment. After carefully revisiting Onuki and Kawasaki’s original paper (OK79), we found that our previous description was not accurate. In OK79, anisotropy was effectively incorporated through wave-number–dependent renormalized transport coefficients. Their renormalization-group equations [Eqs.~(4.24)–(4.25) in OK79] explicitly depend on the angle of the wave vector, and the authors identified a fixed point corresponding to the infinite shear-rate limit. Furthermore, OK79 analyzed Model H, while the present work analyzed Models A and B. Because of these methodological differences, a direct comparison of the scaling behavior is not straightforward.

To clarify this point, in the revised manuscript, we removed the subsection "Comparison with OK79" and added the following paragraph at the end of Sec.4:

"Finally, we comment on the relation between the present analysis and the earlier work by Onuki and Kawasaki (OK79). In OK79, the authors studied Model~H under shear flow by assuming isotropic scaling, while introducing anisotropy through wave-number--dependent renormalized transport coefficients. Their renormalization-group equations [Eqs.~(4.24)--(4.25) in OK79] depend on both amplitude and angle of wave vector, and the resulting fixed point corresponds to the infinite shear-rate limit. In contrast, the present study focuses on Models~A and~B using an explicitly anisotropic-scaling framework. Because the two analyses are based on different theoretical formalisms, their scaling predictions are not directly comparable. A promising direction for future work would be to apply OK79's approach to Models~A and~B, or conversely, to extend the present anisotropic-scaling analysis to Model~H, which could help bridge these complementary frameworks."

Furthermore, we removed the following misleading sentence from the introduction: "Although Onuki and Kawasaki's work in 1979 (OK79) previously applied this method for Model H, their analysis did not incorporate the anisotropy inherent to sheared systems, and the resulting fixed point was unstable under shear. "

---

## Round 1 · List of Changes

1. In the introduction, we removed the following sentence: "Although Onuki and Kawasaki's work in 1979 (OK79) previously applied this method for Model H, their analysis did not incorporate the anisotropy inherent to sheared systems, and the resulting fixed point was unstable under shear.

  2. In Sec. 3, we removed the subsection entitled "Comparison with OK79".

  3. In Sec. 4, we added the following paragraph at the end of the manuscript: "Finally, we comment on the relation between the present analysis and the earlier work by Onuki and Kawasaki (OK79). In OK79, the authors studied Model~H under shear flow by assuming isotropic scaling, while introducing anisotropy through wave-number--dependent renormalized transport coefficients. Their renormalization-group equations [Eqs.~(4.24)--(4.25) in OK79] depend on both amplitude and angle of wave vector, and the resulting fixed point corresponds to the infinite shear-rate limit. In contrast, the present study focuses on Models~A and~B using an explicitly anisotropic-scaling framework. Because the two analyses are based on different theoretical formalisms, their scaling predictions are not directly comparable. A promising direction for future work would be to apply OK79's approach to Models~A and~B, or conversely, to extend the present anisotropic-scaling analysis to Model~H, which could help bridge these complementary frameworks."

---

## Editorial Decision

refereeing_in_preparation